# OpenReview forum: "Learning from Mixed-Quality Data via Exploration-Guided Cognitive Manifolds"
_ICLR.cc/2026/Conference — ICLR 2026 Conference Withdrawn Submission_

### Official Review · Reviewer_Zq44 · 2025-10-29

**Soundness:** 3
**Presentation:** 3
**Contribution:** 2
**Rating:** 4
**Confidence:** 2

**Summary:**

This paper addresses the challenge of learning control policies for satellite battery management systems from mixed-quality data. The authors propose an "exploration-guided cognitive manifold" framework that dynamically weights data samples based on the learner's current "exploration degree" - a metric combining state novelty, policy uncertainty, and temporal progression. The framework includes adaptive data utilization, mixed regularization with behavioral cloning and physics constraints, and an exploration-aware Actor-Critic algorithm. Experiments on simulated satellite environments show modest improvements over baseline methods.

**Strengths:**

1. The paper tackles a practically relevant problem - satellite systems do face challenges with data quality and scarcity. The motivation is clear and the application domain is important for aerospace engineering.

2. The experimental setup considers multiple data quality levels and includes relevant baselines from both online and offline RL. The authors attempt to incorporate domain-specific constraints like orbital periodicity and temperature fluctuations.

3. The writing includes detailed mathematical formulations and the overall presentation is comprehensive, covering multiple aspects of the proposed system.

**Weaknesses:**

1. The core contribution appears to be repackaging existing concepts rather than introducing genuinely novel ideas. The "exploration degree" is essentially uncertainty-based weighting, combining epistemic uncertainty, aleatoric uncertainty, and time-based annealing - all well-established concepts in active learning and curriculum learning literature. The paper doesn't adequately distinguish its contribution from this existing work.

2. The theoretical analysis is weak. The convergence proof (Eq. 20) merely states asymptotic convergence without proof. The sample complexity bound appears to be a standard result with minor modifications. There's no formal characterization of when this approach would outperform simpler alternatives, nor any theoretical justification for why the three-component exploration degree formulation is optimal or even necessary.

3. The paper is unnecessarily complex and verbose. Terms like "cognitive manifolds" and "exploration-guided cognitive matching" obfuscate rather than clarify. The same ideas could be expressed much more simply. The paper reads more like a technical report trying to impress with complexity rather than a scientific contribution with clear innovations.

**Questions:**

1. How does the approach differ fundamentally from uncertainty-weighted importance sampling or curriculum learning methods that already exist in the literature?

2. Why not compare against robust optimization methods or distributionally robust RL approaches that are specifically designed for handling data quality issues?

3. Can you provide ablation studies showing the individual contribution of each component? What happens if you just use the exploration degree for data weighting without the complex regularization framework?

4. The experimental improvements are very small. Have you conducted statistical significance tests to ensure these aren't within noise?

---

### Official Review · Reviewer_XgQt · 2025-10-31

**Soundness:** 1
**Presentation:** 2
**Contribution:** 1
**Rating:** 2
**Confidence:** 2

**Summary:**

The paper proposes an exploration-guided framework for learning from mixed-quality data in satellite multi-battery management. It introduces an exploration score combining state novelty, policy uncertainty, and a temporal schedule. It then uses the score to weight data and to blend three regularizers (behavior cloning, adversarial/value-guided, and physics constraints). Experiments on a custom satellite simulator with four “quality” datasets report gains over PPO/TD3/SAC/BC/CQL on power-tracking accuracy and energy efficiency.

**Strengths:**

1. Learning from mixed-quality data with safety constraints is in general an important problem across all domains.

2. This work builds an end-to-end solution customized for the battery management systems use case.

**Weaknesses:**

1. “Cognitive manifold” remains a rhetorical term that only appears in the title and abstract, and there's no concrete manifold construction, geometry, or representation learning objective defined beyond a scalar exploration score.

2. The paper refers to “state novelty,” “policy/understanding uncertainty,” and a system's “degree of understanding,” but does not provide concrete estimators or algorithms to compute them.

3. The “theoretical properties” section states sample-complexity and constraint-satisfaction expressions without assumptions, theorems, or proofs.

4. The application scope is very narrow. All experiments and most of the narrative are specialized to the satellite multi-battery systems, with no public safety/offline RL benchmarks.

5. The related work under-covers core ML on offline/safe/robust RL, data quality/valuation for RL, and exploration under uncertainty.

**Questions:**

None.

---

### Official Review · Reviewer_ZV9h · 2025-10-31

**Soundness:** 2
**Presentation:** 2
**Contribution:** 2
**Rating:** 4
**Confidence:** 2

**Summary:**

This work addresses the problem of effective data selection in settings with large quantities of low quality data, where it is specifically motivated by the example of battery management in satellites. They introduce a framework constructing cognitive manifolds to learn the understanding of the training model to be used for efficient data allocation suited for the current cognitive state. They present their approach as an exploration problem to capture the models uncertainty and select additional data based off of this. They additionally present experimental results on simulated data demonstrating an improvement over baselines.

**Strengths:**

1. The problem of how to effectively prepare data from low quality sources for effective model training is well motivated, and the idea of framing this problem as being state-dependent is compelling.
2. The paper is very well written and enjoyable to read. Specifically, the figures are very impressive.
3. The empirical results demonstrating a more graceful decline in performance compared to baseline methods as data quality declines is compelling.

**Weaknesses:**

1. One stylistic concern is that this paper seems too specific to the example of satellite battery management. For people not knowledgeable in this specific domain, it is not very clear that this approach can generalize to the many others that suffer from this low quality data problem. In the abstract, the paper aims to present a method to address all safety critical domains, but the set up in section 3 is localized just to their motivating example.
2. The definition of exploration degree in equation (7) is vague and it is unclear what specific statistics denote the novelty of the state, diversity of policy outputs, and the stage characteristics of the learning process. Are these quantities defined or learned?
3. The learnable weight function in (9) is critical to the proposed approach, yet there is no indication of how this value is trained?
4. The work only presents results on synthetic data, where results on real world data seem necessary as the work is posed as having significant practical value.

**Questions:**

1. This work seems very similar to many others works in offline RL that consider data reweighting such as [1,2,3,4]. Can you expand on how your work is novel compared to these?
2. How is E_state, E_policy and E_temporal defined in eq (7)?
3. How is the weight function learned?


[1] Uncertainty-Aware Instance Reweighting for Off-Policy Learning
[2] Conservative and Adaptive Penalty for Model-Based Safe Reinforcement Learning
[3] Uncertainty Weighted Actor-Critic for Offline Reinforcement Learning
[4] Uncertainty-based offline reinforcement learning with diversified q-ensemble

---

### Comment · Area_Chair_qV8C · 2025-11-29

Dear Authors,

Reminder that the response/rebuttal period is soon ending!

Best, AC

---

### Note · Authors · 2025-12-01

I have read and agree with the venue's withdrawal policy on behalf of myself and my co-authors.